# Multi-model simulations of aerosol and ozone radiative forcing due to anthropogenic emission changes during the period 1990-2015

Gunnar Myhre[1], Wenche Aas[2], Ribu Cherian[3], William Collins[4], Greg Faluvegi[5], Mark Flanner[6], Piers Forster[7], Øivind Hodnebrog[1], Zbigniew Klimont[8], Marianne T. Lund[1], Johannes Mülmenstädt[3], Cathrine Lund Myhre[2], Dirk Olivié[9], Michael Prather[10], Johannes Quaas[3], Bjørn H. Samset[1], Jordan L. Schnell[10], Michael Schulz[9],  Drew Shindell[11], Ragnhild B. Skeie[1], Toshihiko Takemura[12], Svetlana Tsyro[9]

[1]Center for International Climate and Environmental Research – Oslo (CICERO), Oslo, Norway

[2]NILU – Norwegian Institute for Air Research, Kjeller, Norway

[3]Institute for Meteorology, Universität Leipzig, Germany

[4]Department of Meteorology, University of Reading, Reading, UK

[5]NASA Goddard Institute for Space Studies and Center for Climate Systems Research, Columbia University, New York, USA

[6]Department of Climate and Space Sciences and Engineering, University of Michigan, Ann Arbor MI, USA.

[7]University of Leeds, Leeds, United Kingdom

[8]International Institute for Applied Systems Analysis (IIASA), Laxenburg, Austria

[9]Norwegian Meteorological Institute, Oslo, Norway

[10]Department of Earth System Science, University of California, Irvine, CA 92697-3100, USA

[11]Nicholas School of the Environment, Duke University, Durham, NC 27708 USA

[12]Kyushu University, Fukuoka, Japan

*Correspondence to*: Gunnar Myhre (gunnar.myhre@cicero.oslo.no)

**Abstract**. Over the past few decades, the geographical distribution of emissions of substances that alter the atmospheric energy balance has changed due to economic growth and air pollution regulations. Here, we show the resulting changes to aerosol and ozone abundances and their radiative forcing, using recently updated emission data for the period 1990-2015, as simulated by seven global atmospheric composition models. The models broadly reproduce large-scale changes in surface aerosol and ozone based on observations (e.g., -1 to -3%/yr in aerosols over the US and Europe). The global mean radiative forcing due to ozone and aerosol changes over the 1990-2015 period increased by +0.17 ±0.08 Wm$^{-2}$, with approximately 1/3 due to ozone. This increase is more strongly positive than

that reported in IPCC AR5. The main reasons for the increased positive radiative forcing of aerosols over this period are the substantial reduction of global mean $SO_2$ emissions, which is stronger in the new emission inventory compared to that used in the IPCC analysis, and higher black carbon emissions.

## 1. Introduction

Over the last decades, global temperature has been forced by a range of both natural and anthropogenic drivers (Schmidt et al., 2014b; Solomon et al., 2011). Relative to the period 1984-1998, which ended with a strong El Niño, the period 1998-2012 saw a reduced rate of global warming. A wide range of studies have discussed possible causes of this slowdown (Fyfe et al., 2016; Marotzke and Forster, 2015; Nieves et al., 2015) including discussions of the temperature trend itself (Karl et al., 2015). A record surface temperature over the instrumental period was however reached in 2014 (Karl et al., 2015) with another new record in 2015. Understanding the reasons behind periods with weaker or stronger temperature changes superimposed on the long-term trend in temperature that is continually forced by increased greenhouse gas concentrations is an integral part of the general study of the climate system.

The Intergovernmental Panel on Climate Change (IPCC) Fifth Assessment Report (AR5) had to rely on a limited number of studies for the 1998-2011 period with regard to radiative forcing of short-lived components (Flato et al., 2013; Myhre et al., 2013b). The short-lived components, notably ozone and atmospheric aerosols, are more difficult to quantify in terms of abundance and radiative forcing through atmospheric measurements than the greenhouse gases with lifetimes in the order of decades or longer. Abundances of short-lived components depend on location of emission, and are inhomogeneously distributed in the atmosphere with variability in time, geographical distribution and altitude.

The short-lived compounds of particular importance in terms of radiative forcing include ozone and atmospheric aerosols. Over the last decades, large changes in regional emissions of ozone and aerosol precursors have occurred, with reductions over the US and Europe in response to air quality controls, and a general increase over South and East Asia (Amann et al., 2013; Crippa et al., 2016; Granier et al., 2011; Klimont et al., 2013). The available emission data for various aerosol types differ in magnitude across regions (Wang et al., 2014b). The net effect of these emission changes in terms of changes in the Earth's radiative balance, is not obvious. In addition to a change in the geographical location of the emissions that emphasizes more chemically active, low-latitude regions; different types of aerosols have different impacts on the radiative balance. Some are purely scattering, while others enhance absorption of solar radiation. They may also affect cloud formation, albedo and lifetime through a range of mechanisms (Boucher et al., 2013; Kaufman et al., 2002). Since the net aerosol forcing is negative (cooling), a reduction in anthropogenic primary aerosol emissions and emissions of aerosol precursors implies a positive forcing over the time period of emission reductions.

The aerosols have a variety of types and composition and involve several different forcing mechanisms, specifically aerosol-radiation interactions (previously denoted direct aerosol effect and semi-direct effect when allowing for rapid adjustments) and aerosol-cloud interactions (Boucher et al., 2013). Their forcing over the industrial era has substantial uncertainties, quantified in terms of a total aerosol forcing of -0.9 (-1.9 to -0.1) W m$^{-2}$ (Boucher et al., 2013). The IPCC AR5 mainly relied on Shindell et al. (2013a) for changes over the last 1-2 decades for the total aerosol forcing,

in addition to one study for the direct aerosol effect based on satellite data (Murphy, 2013). The model studies available for the 2000-2010 period based on the results in Shindell et al. (2013a) were few, compared to what was available for earlier time periods. These studies revealed large regional changes in the aerosol forcing over the last decades, but in terms of global mean changes the values were small in magnitude. The clear sky direct aerosol effect over the period 2000-2012 showed small global mean forcing based on the changes in aerosol abundance from MISR satellite data (Murphy, 2013). The total aerosol forcing over the period 1990-2010 and 2000-2010 in IPCC AR5 was quantified as -0.03 and +0.02 W m$^{-2}$, respectively (Myhre et al., 2013b). Tropospheric ozone forcing was estimated to be +0.03 W m$^{-2}$ over the 1990-2010 period.  Kuhn et al. (2014) simulated a weak direct aerosol effect forcing of +0.06 W m$^{-2}$ over the 1996-2010 period, but with a much stronger forcing of +0.42 Wm$^{-2}$ for the total aerosol effect.

At present aerosol forcing is diagnosed using a wide range of methods, with various degrees of sophistication of the aerosol-radiation and aerosol-cloud interactions included. To span this range and take different approaches into account, we encouraged the modelling groups participating in this study to perform aerosol and ozone forcing simulations over the 1990-2015 period with their standard configuration, but using updated emission inventories and more consistent diagnostics. Here, we present the resulting evolution of aerosol and ozone abundances at the regional level, and the resulting radiative forcing. In particular, the aim is to quantify the recent changes in radiative forcing and how those compare to the values reported in the IPCC AR5.

## 2. Methods

The seven global models participating in the present study are described in Table 1. Participating modelling groups are from the EU project ECLIPSE[1] (Stohl et al. 2015) and those joining an open call for collaborating groups. The model setup to derive forcing varies between the models; from fixed meteorology, to one meteorological year, to fixed sea surface temperatures. All models use identical anthropogenic emission data from ECLIPSE for the 1990 to 2015 period (Klimont et al., 2016; Stohl et al., 2015). Several updates and improvements compared to earlier emission data sets were included in this inventory (Klimont et al., 2016). The ECLIPSE emission data are shown in Figure 1 over the period 1990-2015 and compared to emission data used in Coupled Model Intercomparison Project Phase 5 (CMIP5) and to be used in CMIP6. Supplementary Figure S1 show emission data over Europe and south east Asia, respectively. BC emissions are higher in the ECLIPSE data compared to the CMIP5 data, but with similar trend. For SO$_2$ emission the former has a somewhat larger reduction towards the end of the 1990-2015 period than in the CMIP5 data. For the Community Emissions Data System (CEDS) data for CMIP6, the largest changes to the ECLIPSE data are the more pronounced increase in NOx and OC for the end of the 1990 to 2015 period. The CEDS data will be explored through a large set simulations within CMIP6 (Eyring et al., 2016).

All models simulated the main anthropogenic components sulphate, black carbon (BC) and primary organic aerosols (POA). Further, some models include secondary organic aerosols (SOA) and nitrate. Five of the models simulated

---

[1] Evaluating the Climate and Air Quality Impacts of Short-Lived Pollutants (ECLIPSE); European Union Seventh Framework Programme (FP7/2007-2013) under grant agreement no 282688.

ozone changes over the period. The same offline radiative transfer code used for calculating the radiative forcing for OsloCTM2 was adopted for the atmospheric abundance changes from the EMEP model.

Differences in atmospheric abundances can be large due to different meteorological data sets (up to more than 50% in global mean aerosol burden) (Liu et al., 2007) and surface concentrations can be influenced by interannual variation (making 20 year trends in surface ozone due to climate variability as large as caused by changes in emissions ozone precursors) (Barnes et al., 2016), but differences associated with nudging seem to be small (a few percent) (Sand et al., 2017).

The forcing calculations are quantified at the top of the atmosphere for aerosols and at the tropopause for ozone and follow definitions made in IPCC AR5 (Boucher et al., 2013; Myhre et al., 2013b). The consideration of rapid adjustments associated with aerosols for the various models are described in Table 1.

Radiative forcing is defined as a perturbation relative to a reference state, this can be a flexible year and most common to pre-industrial time (Boucher et al., 2013; Myhre et al., 2013b). All the aerosol and ozone forcings shown here are absolute changes (W m$^{-2}$) relative to the 1990 value of each model. Thus all the plots show forcing starting at 0.0 in 1990.

## 3. Results

### 3.1 Trends in aerosol and ozone

Evaluation of aerosol and chemistry models is a huge topic given the large spatial variability in aerosol and chemical species as well as difficulties associated with sampling issues (Schutgens et al., 2016) and the availability of long term measurements. In this study we restrict the comparison between the models and observations to surface fine mode particular matter which we further show have a similar trend as the total column aerosol optical depth (AOD). In the supplementary material we show comparison of surface ozone between the models used in this study and observations. In addition Supplementary Figure S2 presents trends in the tropospheric column and surface ozone from the models showing much larger difference between surface and column than for aerosols. Whereas the forcing efficiency of aerosols is strongly dependent on the surface reflectance and their position in relation to clouds (Haywood and Shine, 1997) the forcing efficiency for ozone is strongly dependent on altitude and most efficient around tropopause altitude (Forster and Shine, 1997; Lacis et al., 1990; MacIntosh et al., 2016).

Six models simulated changes in annually averaged PM$_{2.5}$ (particulate matter with aerodynamic diameters less than 2.5 µm) over the 1990-2015 period. A model-mean linear trend is fitted and shown as a function of latitude and longitude, see Figure 2a. Regional changes in the model-mean range from 2 to 3%/yr reductions over much of the US and Europe to 1 to 2%/yr increases over much of South and East Asia. The intermodel variation is small, as the models simulate broadly similar geographical patterns. Observations of changes in PM$_{2.5}$ based on the atmospheric networks EMEP (Europe) and IMPROVE (US) are available for selected time periods. The PM$_{2.5}$ trends from observations and model mean results are compared in Table 2. The model results have been derived at the model grid of the observational sites. Over Europe the observed trend is limited to the decade 2000-2010 and is -0.5 %/yr larger (more negative) than the model mean (see Tørseth et al. (2012) for description, site selection, and trend methods). Over the

US we have two decades of PM$_{2.5}$ data, 1998-2008 (Hand et al. (2011), Hand et al. (2014)). We compare with the 2000s decade for consistency with the EMEP comparison, and with the 1989-2008 observations for a longer record. The US record shows that greater % reductions occurred in the second decade, and this is matched by the models simulation. Consistent with the EU record, the observations are -0.2 %/yr more negative than models over either period. Thus our simulation appears to slightly underestimate the reductions in PM$_{2.5}$ over the US and EU. In Figure 2b the AOD at 550 nm is shown as model mean trend in absolute AOD similar to PM$_{2.5}$ in Figure 1a. Maximum reduction in AOD are of 0.30 (absolute AOD) over Europe and maximum increases are 0.25 over East Asia.

Five models simulated surface ozone changes based on the prescribed emissions of precursors including methane. The resulting annual mean surface ozone change (absolute, in ppb) from 1990 to 2015 is shown in Figure S2. The pattern of ozone change is similar among the models, but with some differences in magnitude. The regional changes in surface ozone have many similarities with the surface PM$_{2.5}$ changes (Fig. 2). Surface ozone increases are seen along maritime shipping routes due to increased NOx emissions. Figures S3 and S4 and Table S1 show the surface changes (ppb decade$^{-1}$) from the models compared to observations over the US and EU. Extensive networks of surface ozone measurements, using the full 2,000 or so air quality sites in both the US and EU, are available from 1993 (US) and 1997 (EU) up to the cutoff date of 2013 (see Schnell et al. (2014); Schnell et al. (2015) for networks and methods). These gridded observations identify small-scale variations in the geographic pattern of ozone trends, which is only partially captured in these simulations. Some of the models capture some of the main seasonal shifts (e.g., decrease in summer peak ozone with increase in winter ozone over the eastern US and Europe).

## 3.2 Direct aerosol effect

The total global, annual mean radiative forcing of the change since 1990 in direct aerosol effect is shown in Figure 3, for seven models, together with the estimate given in IPCC AR5. The model mean is very close to the IPCC AR5 value, but the model spread is large. The model mean direct aerosol effect has a positive forcing in the periods 1995-2000 and 2005-2010, with the forcing over the other 5 year periods being negative or consistent with zero.

The model range for the direct aerosol effect due to changes in sulphate concentrations is smaller than that for the total direct aerosol effect, see Figure 4a. The range for sulphate forcing is a factor of two, slightly lower than the model range from other recent multi-model studies (Myhre et al., 2013a). The differences in sulphate burdens between a much larger group of models in IPCC AR5 was greater (Prather et al., 2013). In all of multi-model analyses, differences are not simply proportional to burden because radiative forcing is calculated with different assumptions of optical properties and to the host model for radiative transfer calculations and background fields of important factors such as clouds and surface albedo (Myhre et al., 2013a; Stier et al., 2013). The IPCC AR5 estimate for direct aerosol effect of sulphate was close to zero for the whole 1990-2010 period, whereas the multi-model mean here is around +0.04 Wm$^{-2}$ in year 2010 with further increase to +0.05 Wm$^{-2}$ in 2015. A main reason for this difference is that in the new ECLIPSE emission inventory, global sulphate precursor emissions show stronger reductions for this period than previous estimates. The ECLIPSE SO$_2$ emission change over the 1990-2015 period is about -20%, including international shipping (Klimont et al., 2016; Stohl et al., 2015). Despite the overall positive direct aerosol forcing of

sulphate over the 1990-2015 period from a global reduction of sulphate, it is negative in the intermediate five-year period 2000-2005.

The model-mean global mean radiative forcing of BC direct aerosol effect increases over the 1990-2010 period by +0.07 Wm$^{-2}$ (see Fig. 4b), with values about 20% lower than in IPCC AR5. Between 2010 and 2015 the multi model-mean drops by 25%. The model spread for BC is generally somewhat larger than for sulphate, where differences in the modeled BC vertical profile are the main contributor (Hodnebrog et al., 2014; Samset et al., 2013). The BC emission increases from 1990 to 2015 are 10% in the global sum, but the increase in radiative forcing is relatively larger, and thus BC radiative forcing does not respond linearly to emissions. The forcing efficiency of BC is generally higher over regions of South and East Asia (increasing emissions) than over Europe and US (decreasing emissions), see Haywood and Ramaswamy (1998).

Figures 5a and 5b show the geographical distribution of the multi-model mean 1990-2015 radiative forcing of the direct aerosol effect for sulphate and BC, respectively. Sulphate forcing changes by +1 to +2 W m$^{-2}$ over the southeastern US and central Europe due to reduced abundances; it changes by -0.5 to -1.5 W m$^{-2}$ over most of South and East Asia. In other regions, the changes are minimal. The changes in the direct aerosol effect of BC are smaller in magnitude and opposite in sign: as much as -0.3 W m$^{-2}$ over the US and EU; as much as +0.3 to +1.0 W m$^{-2}$ over a broad region of the northern tropics and sub-tropics from Africa to East Asia. The multi-model direct aerosol effect forcing of POA is very similar to IPCC AR5 over the 1990-2010 period, and generally small in magnitude (Figure 4c). To a small degree, the POA forcing acts to offset the positive forcing from BC and sulphate over the period 1990-2015. SOA are included in a few models with forcing values over the 1990-2015 period generally of smaller magnitudes than POA. Three of the models have nitrate aerosols included, with a large range in the forcing over the period (Figure 4d). The model range in nitrate forcing is presently larger than for other aerosol compounds (Myhre et al., 2013a; Shindell et al., 2013a). The strong nitrate forcing in the GISS model, which is likely too strong (Shindell et al., 2013a), explains the weak and negative total direct aerosol effect found here. On the other hand, NorESM, showing the highest total direct aerosol forcing, is without nitrate aerosols. That model also shows the strongest BC forcing among the models in this study.

### 3.3 Aerosol-cloud interaction and total aerosol effect

A subset of five models were able to diagnose the forcing from aerosol-cloud interaction, with four models having a weak or slightly positive forcing and one model having a large positive forcing, see Figure 6a. In three of the models rapid adjustments associated with aerosol-cloud interactions are simulated (i.e., in IPCC AR5 terms, they simulate an effective radiative forcing, or ERF), whereas in the two models OsloCTM2 and EMEP the RF (changes only to the cloud albedo) was simulated. The differences in direct aerosol effect found here can largely be explained by differences in the individual aerosol components, but a disentangling of aerosol-cloud interaction is more complex and average differences across the models are not readily attributed (Boucher et al., 2013).

The forcing of the total aerosol effect (the combined aerosol-radiation and aerosol-cloud interaction) based on five models, excluding CESM-CAM5 and ECHAM, are shown in Figure 6b. CESM-CAM5 and ECHAM have both direct aerosol effect very close to the model-mean. All five models have a positive total aerosol effect at the end of the 1990-2015 time period, but the magnitudes vary substantially from near zero to +0.2 W m$^{-2}$. The direct aerosol effect causes part of this spread, but the aerosol-cloud interaction is the major cause. Using the ECLIPSE emission data, we find a range similar to earlier studies, from weak to strongly positive total aerosol forcing (Kuhn et al., 2014), but that differs from the assessment of IPCC AR5, which had a negative total aerosol effect. Here, all models show a positive total aerosol forcing with a model-mean of around +0.1 Wm$^{-2}$ ($0.10 \pm 0.08$ W m$^{-2}$ with the uncertainty given as one standard deviation) for the 1990-2015 period. The semi-direct effect of BC and, absorbing OA, is included in the total aerosol effect for all the models, except NorESM. For two of the models (EMEP and OsloCTM2) the semi-direct effect of BC is quantified to be -0.01 and -0.03 W m$^{-2}$ in 2015 and slightly stronger in 2010. These estimates have been derived by the same method as in Hodnebrog et al. (2014); Samset and Myhre (2015). The spatial distribution of the mean multi-model total aerosol forcing from aerosol changes over the 1990-2015 period is shown in Figure 6c. The positive forcing dominates over most regions from a general reduction in the aerosol abundance reaching a maximum of 4.0 W m$^{-2}$ over Europe. Over South and East Asia aerosol increases over the 1990-2015 period have led to a negative forcing of -3.0 W m$^{-2}$.

### 3.4 Ozone forcing

The subset of five models that simulated ozone changes and their resulting radiative forcing all show positive RF over the entire time period. The multi-model mean forcing is twice the IPCC AR5 estimate, see Figure 7. Three models that used fixed meteorology simulate a relatively stable ozone forcing increase, while the other two models show that interannual variability contributed noise to the calculation of this forcing. For the period from 1990 to 2015 the model-mean forcing is +0.06 Wm$^{-2}$, with a model range of the order of 50% around this value.

In addition to the shorter-lived ozone precursors of NOx, CO, and VOC changes in the observed concentration of CH$_4$ is taken into account, except for the EMEP model. The ozone forcing estimate in IPCC AR5 was based on simulations in Stevenson et al. (2013) and for the period after 2005 on the Representative Concentration Pathways 4.5 (RCP45) scenario which has a weaker increase in the forcing than the RCP85 scenario. The stronger ozone forcing in this work compared to IPCC AR5 is likely to be mainly caused by an increase in NOx over the 1990-2010 period that is more than twice that in the emission data used in IPCC AR5, see Figure 1. Changes in CO and VOC are relatively small in the ECLIPSE data and that used for IPCC AR5. The smaller ozone trend from the EMEP model is partly due to their use of a constant CH$_4$ value in the trend calculations. Quantifying the contribution from the various individual ozone precursors is complicated due to non-linearity (Stevenson et al., 2013).

### 4 Summary and conclusions

A suite of models have simulated ozone and aerosol forcing over the 1990-2015 period, using new emission data from the EU project ECLIPSE (Stohl et al., 2015). In areas where there are good and harmonized measurement network

(US and EU), the models generally reproduce observed large scale surface trends in both compounds. Our key findings based on the updated model simulations are stronger positive radiative forcing of aerosols and ozone over the past 25 years than is reported in IPCC AR5. The global average total, multi-model ozone and aerosol forcing over the period 1990 to 2015 is almost +0.2 $Wm^{-2}$. However, uncertainties are large, and the model diversity of aerosol-cloud interaction is especially pronounced. The model range in the direct aerosol effect can be explained by the individual aerosol components and the diversity in modelling these processes. The model range in the forcing of the direct aerosol effect of nitrate aerosols is large and needs further investigations. The model range in the direct aerosol effect of BC is also large, but recent progress on BC lifetime (Samset et al., 2014) and improved understanding of the importance of high resolution modelling for reproducing surface BC measurements (Wang et al., 2014a) are likely to provide more constrained BC forcing estimates in the future. In a similar way, the aerosol-cloud interaction needs observational constraints for reduced model spread. The regional forcing of aerosol changes over the 1990-2015 period is large with maximum values over Europe (+4.0 $Wm^{-2}$) and South East Asia (-3.0 $Wm^{-2}$).

The dominant forcing mechanism over the 1990-2015 period is changes in the well-mixed greenhouse gases (WMGHG). The global mean forcing due to $CO_2$ increased over this period by 0.66 $Wm^{-2}$ and forcing due to other WMGHG rose by 0.16 $Wm^{-2}$ (see Supplementary material for further information of the calculations). Other anthropogenic forcing mechanisms have had negligible overall changes between 1990 and 2015, though; natural forcing of volcanic eruptions and solar irradiance changes have had large changes during the period, see Prather et al. (2013) and particularly their Table AII.1.2. In particular volcanic eruptions cause strong negative forcing on time scale of a few years. The natural forcing due to volcanic and solar irradiance changes was found to be -0.16 (-0.27 to -0.06) $Wm^{-2}$ over the period 1998-2011 (Myhre et al., 2013b). Whereas previous studies indicated almost zero change in forcing of aerosol and ozone change this study shows by using an updated emission inventory and multi-model simulations a forcing 20% of the WMGHG forcing.

**Acknowledgement**

This study benefitted from the Norwegian research council projects #229796 (AeroCom-P3) and the European Union Seventh Framework Programme (FP7/2007-2013) project # 282688. J. L. Schnell was supported by the National Science Foundation's Graduate Research Fellowship Program (DGE-1321846).

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

**Table 1:** Model description.

| Models | Resolution | Fixed-met or fixed-SST | Rapid adjustment | Anthropogenic aerosol components included | References |
|---|---|---|---|---|---|
| **CESM (CAM5, MAM3, MOZART)** | 1.9° x 2.5° L30 | 1982-2001 climatological monthly varying fixed-SSTs and sea-ice | No (direct effect only) | Sulphate, BC, POA, SOA | (Liu et al., 2012; Neale et al., 2010; Wang et al., 2013) |
| **ECHAM6-HAM2** | T63 (1.8X1.8), L31 | Climatological monthly varying fixed-SST and sea ice extent averaged for the period 1979 to 2008. | Included for semi-direct effect, cloud-aerosol interactions on liquid water clouds (no parameterised effects on ice clouds or convective clouds) | Sulphate, BC, POA | (Stevens et al., 2013; Zhang et al., 2012) |
| **EMEP** | 0.5° x 0.5° L20 | 2010 met | Included for semi-direct effect of BC (CESM-CAM4) | Sulphate, nitrate, BC, POA, SOA | (Simpson et al., 2012) |
| **GISS** | 2.0° x 2.5° L40 | 2000 climatological monthly varying fixed-SSTs and sea-ice | Yes | Sulphate, BC, POA, SOA, nitrate (dust also influenced by other anthropogenic aerosols) | (Schmidt et al., 2014a; Shindell et al., 2013b) |
| **NorESM1** | 1.9° x 2.5° L26 | Climatological monthly varying fixed SSTs and sea ice extent | No | Sulphate, BC, POA (SOA included in POA) | (Bentsen et al., 2013; Iversen et al., 2013; Kirkevåg et al., 2013) |

| | | over the 1990-2013 period | | | |
|---|---|---|---|---|---|
| **OsloCTM2** | T42 2.8° x 2.8° L60 | 2010 met | Included for semi-direct effect of BC (CESM-CAM4) | Sulphate, BC, POA, SOA, nitrate | (Myhre et al., 2009; Skeie et al., 2011) |
| **SPRINTARS** | 1.125˚ x 1.125˚ L56 | Climatological monthly varying fixed SSTs and sea ice extent over the 1988-1992 period | Included | Sulphate, BC, POA, SOA | (Takemura et al., 2009; Takemura et al., 2005) |

**Table 2:** Change in PM$_{2.5}$ given in %/yr over Europe and US for observations and multi-model mean. Values in parenthesis are standard deviations of the observed trends. Models have been sampled at the grid points of the network sites. For the models, periods 2000-2010 and 1990-2010 have been used for comparisons with US observations..

|  | # sites | Observations (%/yr) | Mean-models (%/yr) |
|---|---|---|---|
| Europe 2000-2010, based on EMEP network* | 13 | -2.9 (1.5) | -2.4 |
| US 2000-2009, based on IMPROVE network ** | 153 | -2.1 (2.07) | -1.9 |
| US 1989-2009, based on IMPROVE network** | 59 | -1.5 (1.25) | -1.3 |

*Modified from Tørseth et al. (2012) by extending one additional year. Same trend methods are used.

**Adapted from Hand et al. (2011).

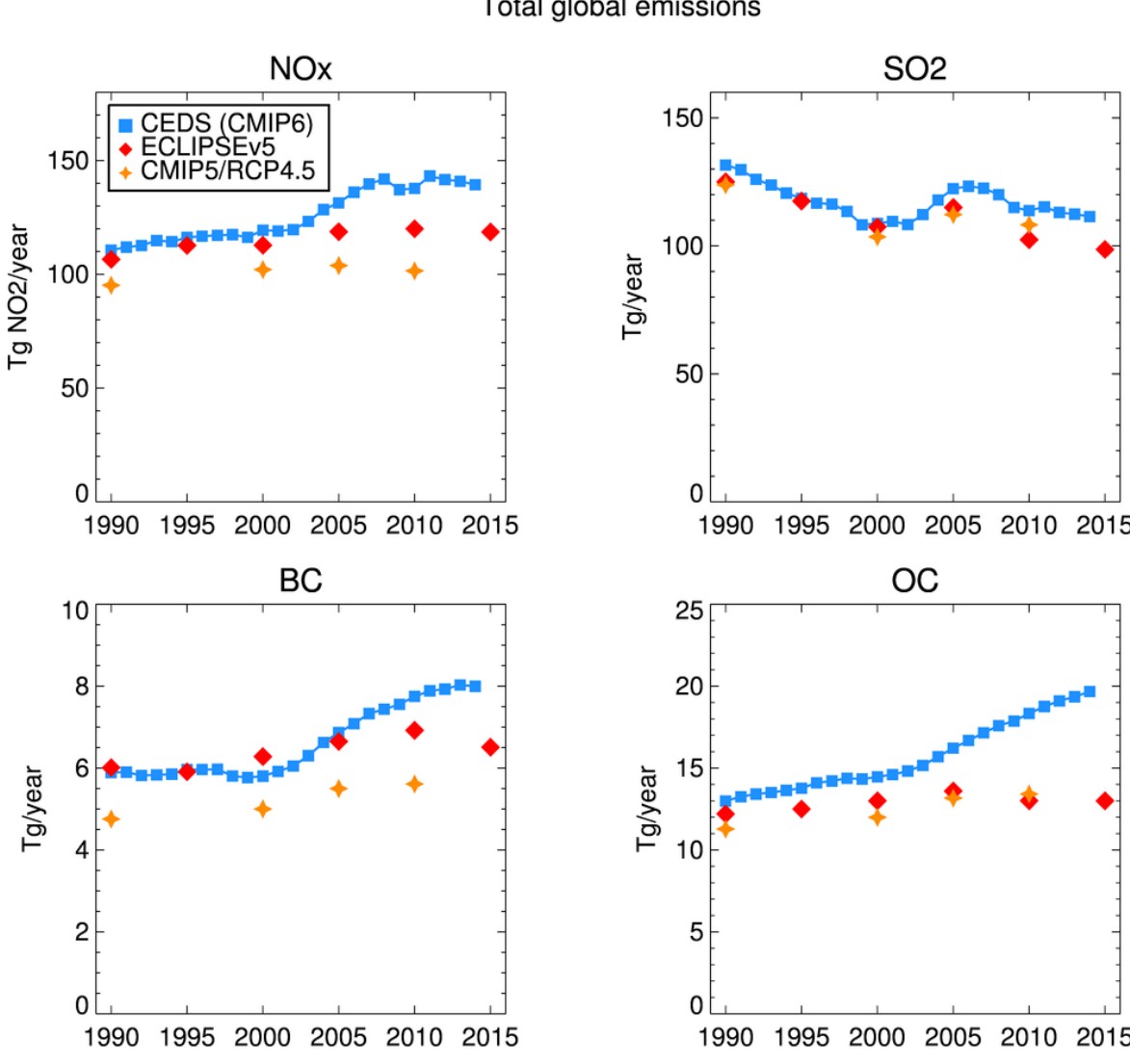

**Figure 1: Global mean emissions for NOx, SO2, BC and OC for ECLIPSE (Klimont et al., 2016), data applied in Coupled Model Intercomparison Project (CMIP5) (Lamarque et al., 2010), and Community Emissions Data System (CEDS) to be used in CMIP6 (Hoesly et al. in preparation) over the period 1990-2015.**

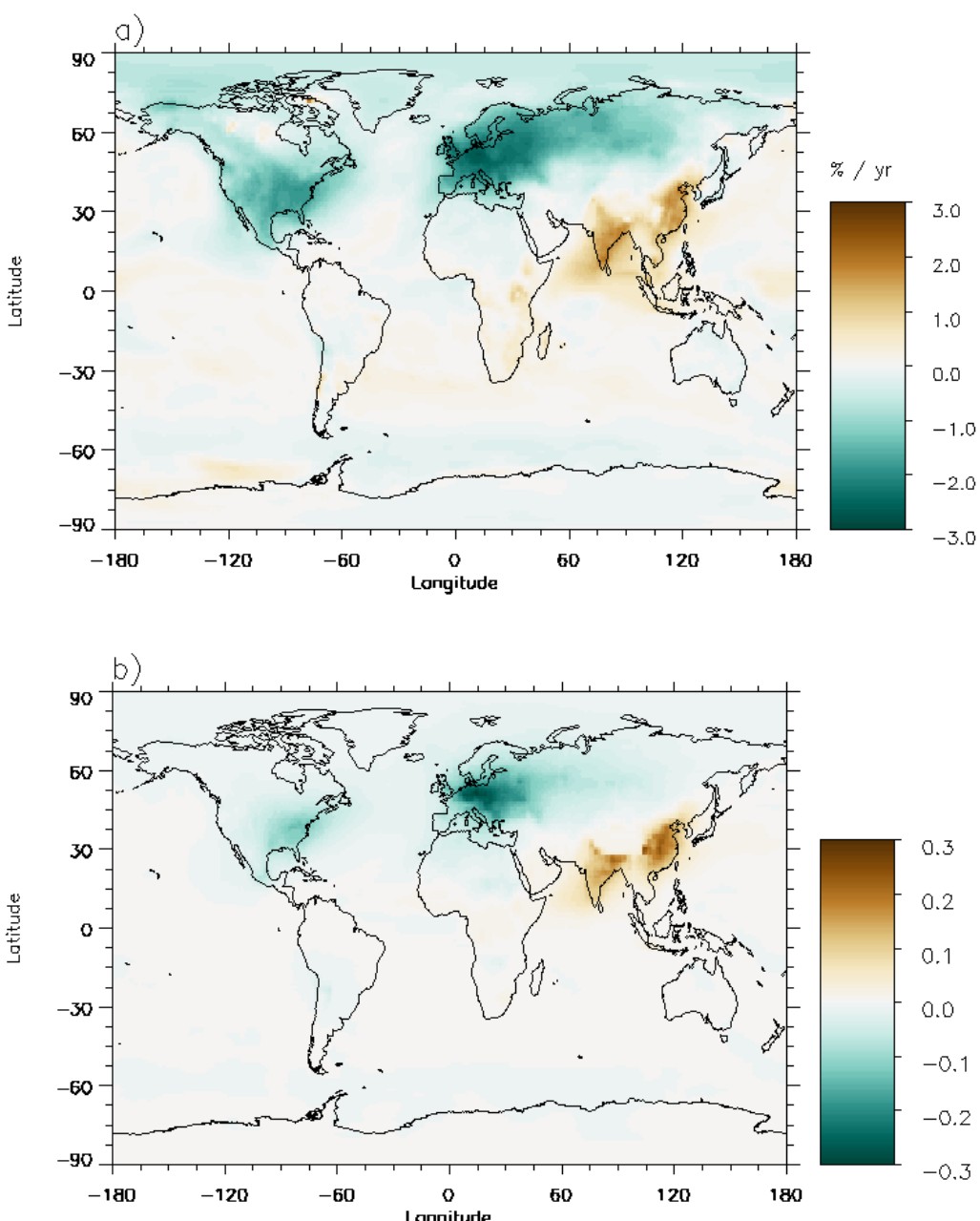

**Figure 2: Multi-model mean linear change in surface PM$_{2.5}$ (a) and aerosol optical depth (AOD) at 550 nm (b), over the 1990-2015 period, simulated by the six models GISS, OsloCTM2, NorESM, CESM-CAM5, EMEP, and SPRINTARS.**

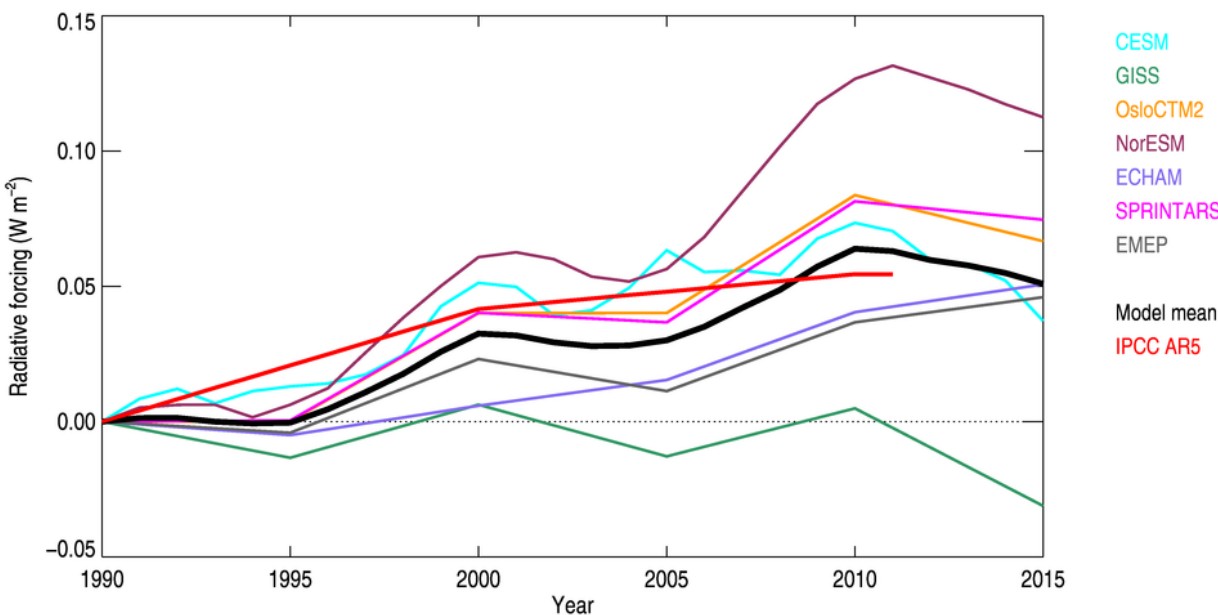

**Figure 3: Radiative forcing (W m⁻²) of the direct aerosol effect over the period 1990-2015 given for seven models (legend lists the models), the multi-model mean is shown in black and the estimate provided in IPCC AR5 is included in red.**

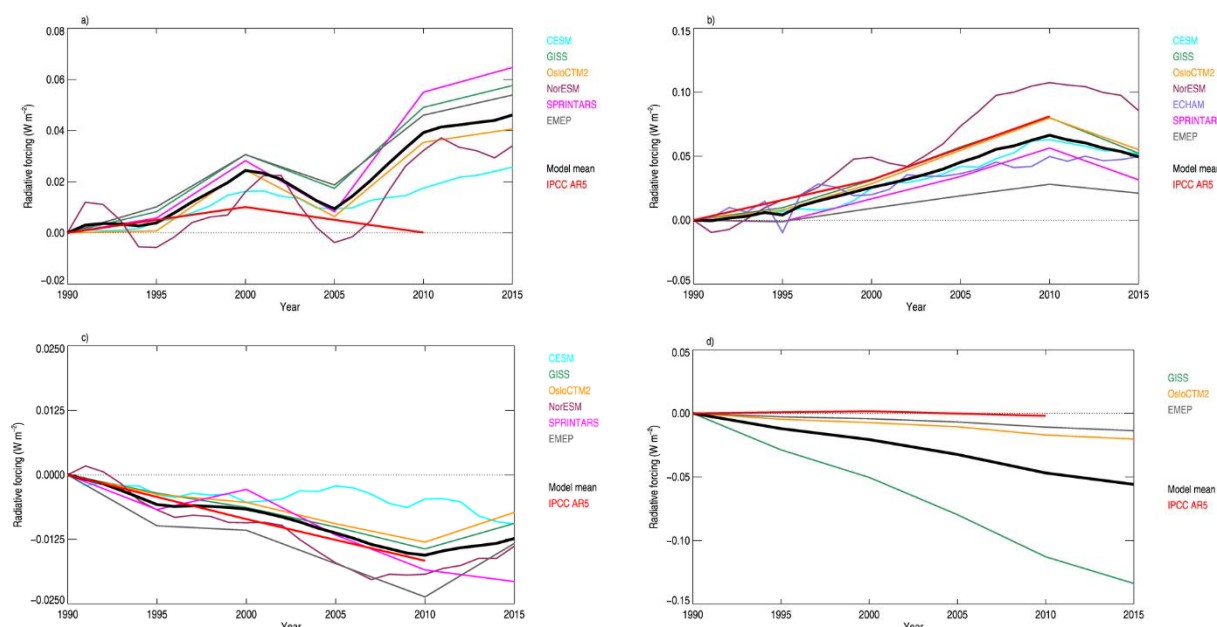

**Figure 4: Radiative forcing (W m⁻²) of the direct aerosol effect by aerosol component (sulphate, a; BC, b; POA, c; nitrate, d) over the period 1990-2015.**

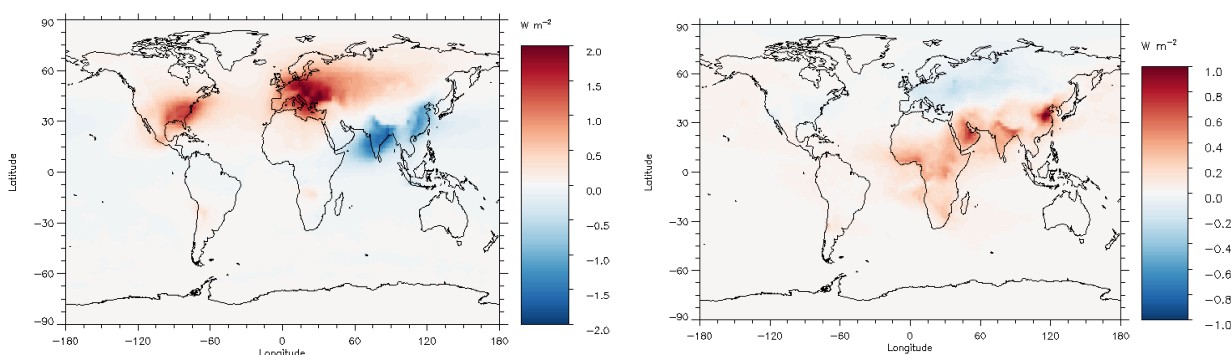

**Figure 5: Geographical distribution of the 1990-2015 radiative forcing (W m$^{-2}$) of the multi-model mean direct aerosol effect sulphate (left) and BC (right) as driven by emission changes.**

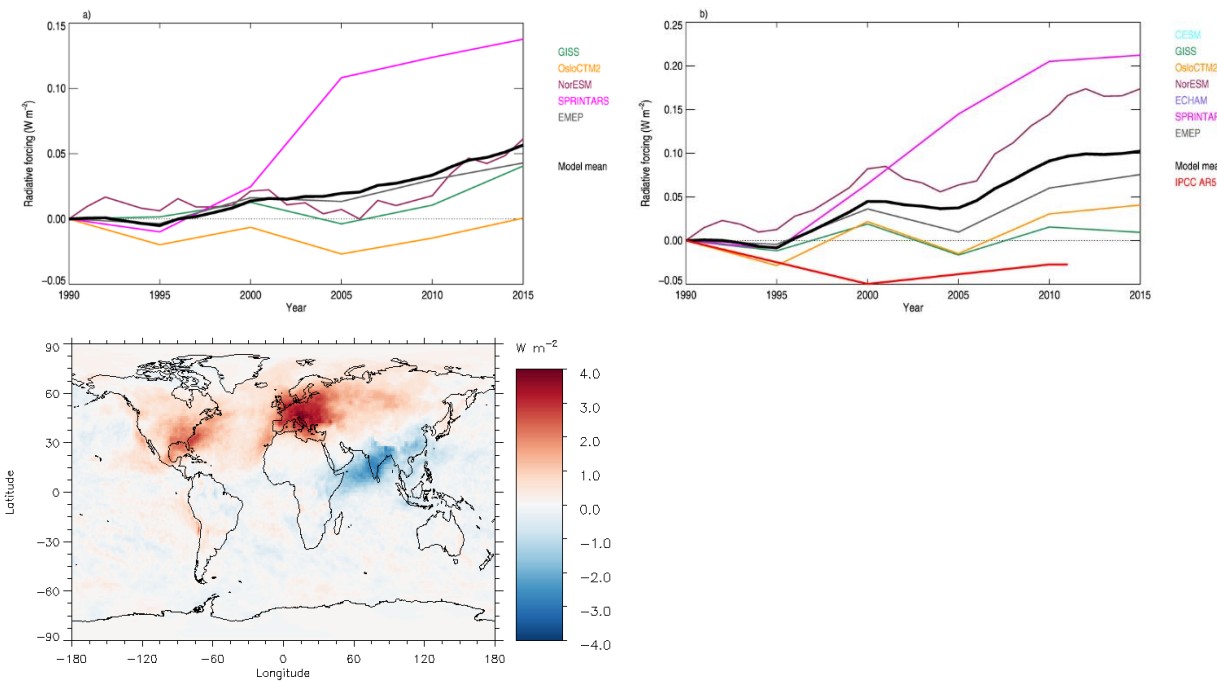

**Figure 6: Radiative forcing (W m⁻²) over the period 1990-2015 of the aerosol-cloud interaction for a subset of the models (a) and total aerosol effect (b). The lower panel shows the geographical distribution of radiative forcing (W m⁻²) of the multi-model mean total aerosol effect.**

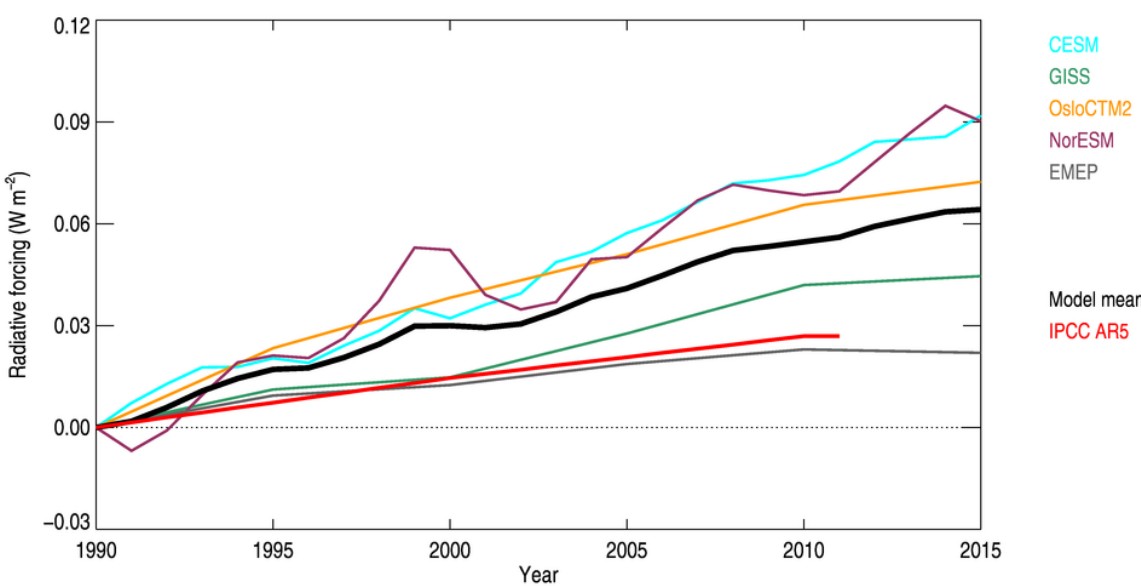

**Figure 7: Radiative forcing (W m$^{-2}$) due to the change in ozone over the period 1990-2015.**