# Peer review of "Multi-model simulations of aerosol and ozone radiative forcing due to anthropogenic emission changes during the period 1990-2015"

_Atmospheric Chemistry and Physics, 2016_

## Referee Comment (RC1) · Anonymous Referee #1 · 5 Sep 2016

This paper summarizes findings from the simulations by 7 models of the period 1990-2015 using updated emissions. The paper succinctly describes the results and discusses the separate roles of aerosol-radiation interactions and aerosol-cloud interactions. The paper is well-written and presents a nice description of the results. I would however strongly suggest that the authors do the following

1) Because so much of the forcing comes from the change in emissions, it would be useful to have a discussion of how those differs from the ACCMIP/RCP projections. Maybe simply trends of major precursors over the region of analysis would be sufficient?

2) The ozone forcing discussion is rather cursory and needs to be extended. Is this simply driven by NOx changes or is methane playing a role, especially over the last 5

years of the analysis period? In addition, why is the IPCC ozone forcing of opposite sign (possibly related to changes in emissions?)

3) It would be useful to put the findings in the overall context of recent forcings (volcanic, solar, stratospheric water vapor, stratospheric ozone).

While this might require an additional simulation, it would be useful to know how much variability in meteorological transport is responsible for the observed/simulated change. In particular, it would be useful to consider using one of the CTMs with a different set of meteorological analysis. Alternatively, the models driven by fixed SSTs could be used with constant emissions (similar to Barnes et al., JGR, 2016) to have a better understanding of the role of internal variability in driving trends over short periods.

––––––––––––––––––––––––––––––

---

## Referee Comment (RC2) · Anonymous Referee #2 · 16 Sep 2016

This study provides an assessment of the evolution of ozone and submicron aerosol atmospheric composition over the 1990-2015 period based on 7 global models and a new emission inventory from the EU ECLIPSE project. This time period is important in global change science for several reasons, including the possible "hiatus" in the global SAT record, and the large changes in regional pollution emissions (decreases in NH mid latitudes and increases in lower latitudes). The study provides global annual average radiative forcing diagnostics and surface concentration changes over the period. The main conclusion is that combined ozone and aerosols changes contributed a net positive global radiative forcing of about +200 mW/m2 between 1990 and 2015. The stronger net positive forcing than that reported in the IPCC AR5 is due to (unexplained) doubling of the ozone forcing, more stringent SO2 reductions and higher BC increases in ECLIPSE, relative to the previous IPCC emission inventory. The paper is clear and

well-written and merits publication in ACP once the following technical issues have been addressed.

1. This study assesses only the effects of anthropogenic emissions changes on the short-lived climate pollutants between 1990-2015. For example, the effects of other global change drivers including physical climate change and land use land cover change are not accounted for in the experimental protocol. The omission of these key drivers may be problematic given that the computed global forcings are quite small. New work from other groups and multimodel assessments is already indicating that physical climate change may be an important driver of chemical changes over this period. At the very least, the title needs to reflect that only changes in anthropogenic pollution emissions are examined and some discussion of the importance of other global change drivers (and why they have or have not been included) needs to be provided to help the readers.

2. A corollary is that the 7 models are based on entirely different chemical and meteorological background states/years (e.g. 2000, 2010 etc.) across the period and this probably represents an important part of the uncertainty ranges, but is not discussed at all. Some discussion and analysis needs to be added to the paper.

3. The paper includes an evaluation of simulated surface concentration trends against observational networks for the period. No measurement data for the entirety of Asia is included in the paper, which is not really acceptable these days, especially because a main focus of the study is on emission changes in Asia.

4. Backing up: Why is this evaluation against surface pollution concentration trends a part of this paper? What is the relationship between surface ozone and aerosol concentrations and their radiative forcings? Please explain. For ozone, the surface concentration change and global forcing changes are rather decoupled. The model/measurement surface ozone trend comparison given is not particularly convincing, and the quantitative details appear to have been relegated to the supplementary

information. Is this poor skill because the models in this study have simplified representation of land-atmosphere interactions? Would it be better for the specific goals of this paper to compare with global column and vertical profile measurements from the satellite records? E.g. MODIS, TES etc. Otherwise, I suggest including "surface concentration trends" in the paper title.

5. Table 1 needs sorting out because inconsistent terminology is used throughout. Please re-design the Table 1 with consistent terminology and acronyms e.g. N/A, 'yes', 'included'. What is L for EMEP? The models that used climatological SSTs and sea ice, for which decade/period? Monthly varying?

6. Table 1 indicates that the GISS model used '2000 met'. If I understand correctly, GISS is a coupled global CCM. There is an option to nudge to reanalysis winds but no full specified dynamics version is available? Please correct here, or provide a published reference for the specified dynamics version of GISS CCM?

7. Is it possible to provide an explanation for the doubled ozone forcing compared to IPCC AR5 value? Is it also due to the updated EU ECLIPSE emissions? The ozone radiative forcing section is very small compared to the aerosol sections! The paper can be improved and more interesting by presenting the major precursor drivers of the changes, and the reasons for discrepancies with other results.

8. I read several times over, and I find it difficult to understand exactly what is included in the multi-model mean "total aerosol forcing"? Can this definition be made clearer? I realize it is challenging in multimodel studies when models simulate different aerosol types and some represent aerosol-cloud interactions while others do not.

9. The uncertainty range needs to be added to the total forcing of +200mW/m2 in the abstract.

10. Would it be useful to add a comparison to the total CO2 forcing across this period? I believe the SLCP forcing is about 40% of the CO2 forcing across the period.

11. Page 4, Line 20 states: "Five models simulated surface ozone changes based on the prescribed emissions of precursors including methane." Does this mean that the models all have chemically dynamic ("flux-based") full methane cycle simulations? Or do the models prescribe methane atmospheric concentrations based on observed amounts? Should methane radiative forcings be included in this analysis? If the models are using flux-based methane simulations then more information is needed about the natural emissions and some solid evaluation of the simulated methane concentrations.

---

## Referee Comment (RC3) · Anonymous Referee #3 · 7 Oct 2016

In their manuscript "Multi-model simulations of aerosol and ozone radiative forcing for the period 1990-2015", the authors present an updated multi-model estimate of aerosol and ozone radiative forcing for the period 1990 to 2015.

Using an updated emission dataset, aerosol forcing is found to be stronger positive in the new estimate as compared to IPCC AR5 assessment report, which is attributed to a stronger decline of SO2 emissions and stronger BC emissions in the updated inventory. The manuscript is well written and the results are valid. Yet, given the differences in the used emissions and the selection of models, the results are not particularly surprising, which left me a bit confused about the overall scientific objectives of this work. Before recommending this work for publication, I would encourage the authors to emphasize the objectives of their work as well as to deepen the analysis of the process chain leading to the simulated forcing changes.

[Figure]

**General issues**

- As a whole, I was missing a more in-depth analysis and discussion of how much of the differences to prior scenarios are simply a reflection of emission changes (in terms of magnitude and location) or due to an arbitrary selection of models (which also have changed from their state at AR5).

- As emission changes seem to drive most of the simulated changes, the assumptions leading to these differences should be discussed in some detail. At present, the readers are simply referred to Stohl et al. (2015) but given the importance of these changes it should be possible to repeat the key points here.

- You fit a linear trend to emissions changes, how good (or bad) is this assumption? Are all local emissions changes at least monotonic?

- The use of "forcing" is at times a bit ambiguous. Readers are used to forcing with respect to the pre-industrial period, however you seem to use it for the period of interest. Maybe this could be made more clear?

**Specific issues**

- *Page 5: "In all of multi-model analyses, differences are not simply proportional to burden because radiative forcing is calculated with different assumptions of optical properties and the radiative transfer calculations."* Forcing depends on more factors than just burden, optical properties and radiative transfer, as has been quantified in recent AeroCom experiments. Please expand on this.

[Figure]

- *Page 5: "BC is generally a more efficient absorber over regions of South and East Asia (increasing emissions) than over Europe and US"* This is ambiguous. Pure BC is likely to be of similar absorption efficiency. Do you mean BC has different coatings (or coating thickness) or do you mean it has higher forcing efficiency (due to surface albedo differences or changes in cloud cover)?

- Table 1: The caption needs to unambiguously describe the table. Acronyms like SOA vs. POM are not clear here.

- Table 2: It is entirely unclear how the comparison was done. Have models been sub-sampled at measurement sites / times to minimize sampling errors?

- Figure 1: The caption is not clear: this should be specific that this is a linear fit etc. . .

---

## Author Comment (AC1) · 24 Nov 2016

We thank the three Reviewers for positive and constructive comments. Our responses are given in **bold** font and changes to the manuscript in red.

Anonymous Referee #1

This paper summarizes findings from the simulations by 7 models of the period 1990- 2015 using updated emissions. The paper succinctly describes the results and discusses the separate roles of aerosol-radiation interactions and aerosol-cloud interactions. The paper is well-written and presents a nice description of the results. I would however strongly suggest that the authors do the following

1) Because so much of the forcing comes from the change in emissions, it would be useful to have a discussion of how those differs from the ACCMIP/RCP projections. Maybe simply trends of major precursors over the region of analysis would be sufficient?

   **Response: A figure comparing the Eclipse emission data with CMIP5/ACCMIP/RCP and the new CEDS data (for use in CMIP6) is included in the revised manuscript. Two figures on the regional emissions are added to the supplementary. A description of the figure including a reference to the ACPD paper by Klimont et al. have been added the manuscript as follows:**

   'The ECLIPSE emission data are shown in Figure 1 over the period 1990-2015 and compared to emission data used in Coupled Model Intercomparison Project (CMIP5) and to be used in CMIP6. Supplementary Figure S1 show emission data over Europe and south east Asia, respectively. BC emissions are higher in the ECLIPSE data compared to the CMIP5 data, but with similar trend. For $SO_2$ emission the former has a somewhat larger reduction towards the end of the 1990-2015 period than in the CMIP5 data. For the Community Emissions Data System (CEDS) data for CMIP6, the largest changes to the ECLIPSE data are the more pronounced increase in NOx and OC for the end of the 1990 to 2015 period. The CEDS data will be explored through a large set simulations within CMIP6 (Eyring et al., 2016).'

2) The ozone forcing discussion is rather cursory and needs to be extended. Is this simply driven by NOx changes or is methane playing a role, especially over the last 5 years of the analysis period? In addition, why is the IPCC ozone forcing of opposite sign (possibly related to changes in emissions?)

   **Response: The ozone forcing discussion is extended including the role of methane. The IPCC forcing is weaker than found in our work, but of same sign. The end value in 2011 for the IPCC data is assumed to be the same as in 2010. The following text has been added to the manuscript:**

   'In addition to the shorter-lived ozone precursors of NOx, CO, and VOC changes in the concentration of $CH_4$ is taken into account, except for the EMEP model. The ozone forcing estimate in IPCC AR5 was based on simulations in Stevenson et al. (2013) and for the period after 2005 on the Representative Concentration Pathways 4.5 (RCP45) scenario which has a weaker increase in the forcing than the RCP85 scenario. Smaller ozone trend from the EMEP model is partly due to that a constant $CH_4$ value used in the trend calculations.'

3) It would be useful to put the findings in the overall context of recent forcings (volcanic, solar, stratospheric water vapor, stratospheric ozone).

**Response: A new paragraph has been included to put the aerosol and ozone forcing in context of other anthropogenic forcings and the two natural forcings of solar irradiance and volcanic eruption. The following paragraph is added to the manuscript:**

'The dominant forcing mechanism over the 1990-2015 period is changes in the well-mixed greenhouse gases (WMGHG). The global mean forcing due to $CO_2$ increased over this period by 0.66 $Wm^{-2}$ and forcing due to other WMGHG rose by 0.16 $Wm^{-2}$ (see Supplementary material for further information of the calculations). Other anthropogenic forcing mechanisms have had negligible overall changes between 1990 and 2015, though; natural forcing of volcanic eruptions and solar irradiance changes have had large changes during the period, see Prather et al. (2013) and particularly their Table AII.1.2. Whereas previous studies indicated almost zero change in forcing of aerosol and ozone change this study shows by using an updated emission inventory and multi-model simulations a forcing 20% of the WMGHG forcing.'

While this might require an additional simulation, it would be useful to know how much variability in meteorological transport is responsible for the observed/simulated change. In particular, it would be useful to consider using one of the CTMs with a different set of meteorological analysis. Alternatively, the models driven by fixed SSTs could be used with constant emissions (similar to Barnes et al., JGR, 2016) to have a better understanding of the role of internal variability in driving trends over short periods.

**Response: We have investigated in an earlier study that interannual variation using consistent meteorological fields provide small differences (http://www.atmos-chem-phys.net/11/11293/2011/acp-11-11293-2011-discussion.html). However, differences between various meteorological data sets give larger differences. We have added the following sentences to the manuscript:**

'Differences in atmospheric abundances can be large due to different meteorological data sets (Liu et al., 2007) and surface concentrations can be influenced by interannual variation (Barnes et al., 2016), but differences associated with nudging seem to be small (Sand et al., in preparations). '

Anonymous Referee #2

This study provides an assessment of the evolution of ozone and submicron aerosol atmospheric composition over the 1990-2015 period based on 7 global models and a new emission inventory from the EU ECLIPSE project. This time period is important in global change science for several reasons, including the possible "hiatus" in the global SAT record, and the large changes in regional pollution emissions (decreases in NH mid latitudes and increases in lower latitudes). The study provides global annual average radiative forcing diagnostics and surface concentration changes over the period. The main conclusion is that combined ozone and aerosols changes contributed a net positive global radiative forcing of about +200 mW/m2 between 1990 and 2015. The stronger net positive forcing than that reported in the IPCC AR5 is due to (unexplained) doubling of the ozone forcing, more stringent SO2 reductions and higher BC increases in ECLIPSE, relative to the previous IPCC emission inventory. The paper is clear and well-written and merits publication in ACP once the following technical issues have been addressed.

1. This study assesses only the effects of anthropogenic emissions changes on the short-lived climate pollutants between 1990-2015. For example, the effects of other global change drivers including physical climate change and land use land cover change are not accounted for in the experimental protocol. The omission of these key drivers may be problematic given that the

computed global forcings are quite small. New work from other groups and multimodel assessments is already indicating that physical climate change may be an important driver of chemical changes over this period. At the very least, the title needs to reflect that only changes in anthropogenic pollution emissions are examined and some discussion of the importance of other global change drivers (and why they have or have not been included) needs to be provided to help the readers.

**Response: A new paragraph included in the summary and the title has been changed. The added paragraph reads as follows:**

'The dominant forcing mechanism over the 1990-2015 period is changes in the well-mixed greenhouse gases (WMGHG). The global mean forcing due to $CO_2$ increased over this period by 0.66 $Wm^{-2}$ and forcing due to other WMGHG rose by 0.16 $Wm^{-2}$ (see Supplementary material for further information of the calculations). Other anthropogenic forcing mechanisms have had negligible overall changes between 1990 and 2015, though; natural forcing of volcanic eruptions and solar irradiance changes have had large changes during the period, see Prather et al. (2013) and particularly their Table AII.1.2. Whereas previous studies indicated almost zero change in forcing of aerosol and ozone change this study shows by using an updated emission inventory and multi-model simulations a forcing 20% of the WMGHG forcing.'

2. A corollary is that the 7 models are based on entirely different chemical and meteorological background states/years (e.g. 2000, 2010 etc.) across the period and this probably represents an important part of the uncertainty ranges, but is not discussed at all. Some discussion and analysis needs to be added to the paper.

**Response: See response to Reviewer 1. The following change to the manuscript has been done:**
'Differences in atmospheric abundances can be large due to different meteorological data sets (Liu et al., 2007) and surface concentrations can be influenced by interannual variation (Barnes et al., 2016), but differences associated with nudging seem to be small (Sand et al., in preparations). '

3. The paper includes an evaluation of simulated surface concentration trends against observational networks for the period. No measurement data for the entirety of Asia is included in the paper, which is not really acceptable these days, especially because a main focus of the study is on emission changes in Asia.

**Response: Unfortunately there are, as far as the authors are aware of, no regional networks in Asia with long term measurement of PM2.5 back to 2000 to be compared with the observed trends in Europe and North America as presented in table 2. Most of the sites with long term measurements are situated in cities and these are difficult to use to assess regional trends. However, the resent years there has been established regional networks and sites which can be used for studying temporal and spatial variability of aerosols in Asia. I.e. Wang et (2015) present PM10, PM2.5 and PM1 data from 24 CAWNET (China Atmosphere Watch Network) sites, whereof 11 are regional or remote sites with measurements from 2006 to 2014.  The trends presented here does however not conclude on any large scale regional trends in China in this period. Trends in surface sulfate concentration and aerosol optical depth is available for a few site in Asia, but this work would need proper documentation before comparing to the model data in this study. See response to next comment on changes made to the manuscript.**

**Wang Y. Q., Zhang X. Y., Sun J. Y., Zhang X. C., Che H. Z., Li Y. Spatial and temporal variations of the concentrations of PM10, PM2.5 and PM1 in China. Atmos. Chem. Phys. 2015; 15: 13585-13598.**

4. Backing up: Why is this evaluation against surface pollution concentration trends a part of this paper? What is the relationship between surface ozone and aerosol concentrations and their radiative forcings? Please explain. For ozone, the surface concentration change and global forcing changes are rather decoupled. The model/measurement surface ozone trend comparison given is not particularly convincing, and the quantitative details appear to have been relegated to the supplementary information. Is this poor skill because the models in this study have simplified representation of land-atmosphere interactions? Would it be better for the specific goals of this paper to compare with global column and vertical profile measurements from the satellite records? E.g. MODIS, TES etc. Otherwise, I suggest including "surface concentration trends" in the paper title.

   **Response: We have added an explanatory paragraph in beginning of the result section as the following:**
   'Evaluation of aerosol and chemistry models is a huge topic given the large spatial variability in aerosol and chemical species as well as difficulties associated with sampling issues (Schutgens et al., 2016) and the availability of long term measurements. In this study we restrict the comparison between the models and observations to surface fine mode particular matter which we further show have a similar trend as the total column aerosol optical depth (AOD). In the supplementary material we show comparison of surface ozone between the models used in this study and observations. In addition Supplementary Figure S2 presents trends in the tropospheric column and surface ozone from the models showing much larger difference between surface and column than for aerosols. Whereas the forcing efficiency of aerosols is strongly dependent on the surface reflectance and their position in relation to clouds (Haywood and Shine, 1997) the forcing efficiency for ozone is strongly dependent on altitude and most efficient around tropopause altitude (Forster and Shine, 1997; Lacis et al., 1990; MacIntosh et al., 2016).'

5. Table 1 needs sorting out because inconsistent terminology is used throughout. Please re-design the Table 1 with consistent terminology and acronyms e.g. N/A, 'yes', 'included'. What is L for EMEP? The models that used climatological SSTs and sea ice, for which decade/period? Monthly varying?

   **Response: We have included number of layers (L) for the EMEP model and used a consistent terminology.**

6. Table 1 indicates that the GISS model used '2000 met'. If I understand correctly, GISS is a coupled global CCM. There is an option to nudge to reanalysis winds but no full specified dynamics version is available? Please correct here, or provide a published reference for the specified dynamics version of GISS CCM?

   **Response: The table has been updated with the following information for the GISS model:**
   '2000 climatological monthly varying fixed-SSTs and sea-ice'

7. Is it possible to provide an explanation for the doubled ozone forcing compared to IPCC AR5 value? Is it also due to the updated EU ECLIPSE emissions? The ozone radiative forcing section is very small compared to the aerosol sections! The paper can be improved and more interesting by presenting the major precursor drivers of the changes, and the reasons for discrepancies with other results.

**Response: We have indicated that the RCP85 simulations had higher ozone forcing than RCP45 which was used in IPCC AR5. The following is added to the manuscript in section 3.4:**

'In addition to the shorter-lived ozone precursors of NOx, CO, and VOC changes in the concentration of $CH_4$ is taken into account, except for the EMEP model. The ozone forcing estimate in IPCC AR5 was based on simulations in Stevenson et al. (2013) and for the period after 2005 on the Representative Concentration Pathways 4.5 (RCP45) scenario which has a weaker increase in the forcing than the RCP85 scenario. Smaller ozone trend from the EMEP model is partly due to that a constant $CH_4$ value used in the trend calculations.'

8. I read several times over, and I find it difficult to understand exactly what is included in the multi-model mean "total aerosol forcing"? Can this definition be made clearer? I realize it is challenging in multimodel studies when models simulate different aerosol types and some represent aerosol-cloud interactions while others do not.

   **Response: We have defined the how we use the term total aerosol effect. The following is added to the manuscript:**
   '(the combined aerosol-radiation and aerosol-cloud interaction)'

9. The uncertainty range needs to be added to the total forcing of +200mW/m2 in the abstract.

   **Response: Uncertainty range added in the abstract.**

10. Would it be useful to add a comparison to the total CO2 forcing across this period? I believe the SLCP forcing is about 40% of the CO2 forcing across the period.

    **Response: A discussion is included in a paragraph in the summary and reads as follows:**
    'The dominant forcing mechanism over the 1990-2015 period is changes in the well-mixed greenhouse gases (WMGHG). The global mean forcing due to $CO_2$ increased over this period by 0.66 $Wm^{-2}$ and forcing due to other WMGHG rose by 0.16 $Wm^{-2}$ (see Supplementary material for further information of the calculations). Other anthropogenic forcing mechanisms have had negligible overall changes between 1990 and 2015, though; natural forcing of volcanic eruptions and solar irradiance changes have had large changes during the period, see Prather et al. (2013) and particularly their Table AII.1.2. Whereas previous studies indicated almost zero change in forcing of aerosol and ozone change this study shows by using an updated emission inventory and multi-model simulations a forcing 20% of the WMGHG forcing.'

11. Page 4, Line 20 states: "Five models simulated surface ozone changes based on the prescribed emissions of precursors including methane." Does this mean that the models all have chemically dynamic ("flux-based") full methane cycle simulations? Or do the models prescribe methane atmospheric concentrations based on observed amounts? Should methane radiative forcings be included in this analysis? If the models are using flux-based methane simulations then more information is needed about the natural emissions and some solid evaluation of the simulated methane concentrations.

    **Response: We have added that changes in $CH_4$ concentrations have be included, except for the EMEP model.**

Anonymous Referee #3

In their manuscript "Multi-model simulations of aerosol and ozone radiative forcing for the period 1990-2015", the authors present an updated multi-model estimate of aerosol and ozone radiative forcing for the period 1990 to 2015.
Using an updated emission dataset, aerosol forcing is found to be stronger positive in the new estimate as compared to IPCC AR5 assessment report, which is attributed to a stronger decline of SO2 emissions and stronger BC emissions in the updated inventory. The manuscript is well written and the results are valid. Yet, given the differences in the used emissions and the selection of models, the results are not particularly surprising, which left me a bit confused about the overall scientific objectives of this work. Before recommending this work for publication, I would encourage the authors to emphasize the objectives of their work as well as to deepen the analysis of the process chain leading to the simulated forcing changes.

**General issues**
• As a whole, I was missing a more in-depth analysis and discussion of how much of the differences to prior scenarios are simply a reflection of emission changes (in terms of magnitude and location) or due to an arbitrary selection of models (which also have changed from their state at AR5).
**Response: We have added a figure in the manuscript as well as two figures for the supplementary material. Text associated with the figures are added as follows:**
 'The ECLIPSE emission data are shown in Figure 1 over the period 1990-2015 and compared to emission data used in Coupled Model Intercomparison Project (CMIP5) and to be used in CMIP6. Supplementary Figure S1 show emission data over Europe and south east Asia, respectively. BC emissions are higher in the ECLIPSE data compared to the CMIP5 data, but with similar trend. For $SO_2$ emission the former has a somewhat larger reduction towards the end of the 1990-2015 period than in the CMIP5 data. For the Community Emissions Data System (CEDS) data for CMIP6, the largest changes to the ECLIPSE data are the more pronounced increase in NOx and OC for the end of the 1990 to 2015 period. The CEDS data will be explored through a large set simulations within CMIP6 (Eyring et al., 2016).'

• As emission changes seem to drive most of the simulated changes, the assumptions leading to these differences should be discussed in some detail. At present, the readers are simply referred to Stohl et al. (2015) but given the importance of these changes it should be possible to repeat the key points here.
**Response: We have added a figure in the manuscript as well as two figures for the supplementary material. Text associated with the figures are added and see response to the previous comment.**

• You fit a linear trend to emissions changes, how good (or bad) is this assumption? Are all local emissions changes at least monotonic?
**Response: We have compared trends in the concentration and not in the emissions.**

• The use of "forcing" is at times a bit ambiguous. Readers are used to forcing with respect to the pre-industrial period, however you seem to use it for the period of interest. Maybe this could be made more clear?
**Response: We have highlighted the point on time period. The following changes have been implemented:**
'Radiative forcing is defined as a perturbation relative to a reference state, this can be a flexible year and most common to pre-industrial time (Boucher et al., 2013; Myhre et al., 2013b). All the aerosol and ozone forcings shown here are absolute changes (W m$^{-2}$) relative to the 1990 value of each model. Thus all the plots show forcing starting at 0.0 in 1990.'

**Specific issues**

• Page 5: "In all of multi-model analyses, differences are not simply proportional to burden because radiative forcing is calculated with different assumptions of optical properties and the radiative transfer calculations." Forcing depends on more factors than just burden, optical properties and radiative transfer, as has been quantified in recent AeroCom experiments. Please expand on this.

**Response: The following is added:**

**'to the host model for radiative transfer calculations and background fields of important factors such as clouds and surface albedo (Myhre et al., 2013a; Stier et al., 2013)'**

• Page 5: "BC is generally a more efficient absorber over regions of South and East Asia (increasing emissions) than over Europe and US" This is ambiguous. Pure BC is likely to be of similar absorption efficiency. Do you mean BC has different coatings (or coating thickness) or do you mean it has higher forcing efficiency (due to surface albedo differences or changes in cloud cover)?

**Response: We have corrected this to state that the forcing efficiency is higher.**

• Table 1: The caption needs to unambiguously describe the table. Acronyms like SOA vs. POM are not clear here.

**Response: We have modified the table and now used POA (primary OA) and SOA consistently**

• Table 2: It is entirely unclear how the comparison was done. Have models been sub-sampled at measurement sites / times to minimize sampling errors?

**Response: The following added to the table caption:**

**'Models have been sampled at the grid points of the network sites and for the US periods 2000-2010 and 1990-2010 have been derived. '**

• Figure 1: The caption is not clear: this should be specific that this is a inear fit etc: : :

**Response: It is added 'linear'**

---

## Author Response (AR2)

**Response to Editor comments and track change version of the manuscript**

**Comments to the Author:**
I thought that the response to comments document was unclear on several points. I asked the 3 reviewers to review the paper again, and unfortunately, all 3 reviewers declined to review again. My sense is that the paper is good, but that you were perhaps sloppy in your responses to the reviews. To avoid delay, rather than sending the paper to a new set of reviewers, I have decided that I will follow up with you on several points that I thought were unclear. Please make another attempt to clarify these points.

**Response:** We thank the Editor for the careful reading of our revised manuscript and response. We agree that some of the points could have be clarified better. The new parts are given in *italic* text within the yellow boxes. However, we feel that some of the comments would need a community effort and would take long time to resolve and delaying the manuscript substantially.

Reviewer #1

Comment 2 - You compare RCP4.5 with RCP8.5, but it is not clear how these relate with the current paper. Please rewrite the explanation of why ozone forcing is different in this study.

**Response**: We agree that this explanation could be further improved. We have added a sentence on NOx emissions which is higher in the ECLIPSE data compared to data used for IPCC AR5.

**Original change in text:** 'In addition to the shorter-lived ozone precursors of NOx, CO, and VOC changes in the concentration of $CH_4$ is taken into account, except for the EMEP model. The ozone forcing estimate in IPCC AR5 was based on simulations in Stevenson et al. (2013) and for the period after 2005 on the Representative Concentration Pathways 4.5 (RCP45) scenario which has a weaker increase in the forcing than the RCP85 scenario. Smaller ozone trend from the EMEP model is partly due to that a constant $CH_4$ value used in the trend calculations.'

**New change in text:** 'In addition to the shorter-lived ozone precursors of NOx, CO, and VOC changes in the concentration of $CH_4$ is taken into account, except for the EMEP model. The ozone forcing estimate in IPCC AR5 was based on simulations in Stevenson et al. (2013) and for the period after 2005 on the Representative Concentration Pathways 4.5 (RCP45) scenario which has a weaker increase in the forcing than the RCP85 scenario. *The stronger ozone forcing in this work compared to IPCC AR5 is likely to be mainly caused by an increase in NOx over the 1990-2010 period that is more than twice that in the emission data used in IPCC AR5, see Figure 1. Changes in CO and VOC are relatively small in the ECLIPSE data and that used for IPCC AR5. The smaller* ozone trend from the EMEP model is partly due to *their use of* a constant $CH_4$ value in the trend calculations. *Quantifying the contribution from the various individual ozone precursors is complicated due to non-linearity (Stevenson et al., 2013).*'

Comment 3 - The response is good, but the reviewer had asked that natural forcings be compared with those calculated here. You mention "large changes" in natural forcings, but do not provide a quantitative comparison. Please rewrite so that this comparison is made quantitatively.

**Response**: We have added two more sentences describing the natural forcing, including quantification of the forcing as performed in IPCC AR5. Even no major volcanic eruption took place the volcanic forcing has been of importance during the investigated period.

**Original change in text:** 'The dominant forcing mechanism over the 1990-2015 period is changes in the well-mixed greenhouse gases (WMGHG). The global mean forcing due to $CO_2$ increased over this period by 0.66 Wm$^{-2}$ and forcing due to other WMGHG rose by 0.16 Wm$^{-2}$ (see Supplementary material for further information of the calculations). Other anthropogenic forcing mechanisms have had negligible overall changes between 1990 and 2015, though; natural forcing of volcanic eruptions and solar irradiance changes have had large changes during the period, see Prather et al. (2013) and particularly their Table AII.1.2. Whereas previous studies indicated almost zero change in forcing of aerosol and ozone change this study shows by using an updated emission inventory and multi-model simulations a forcing 20% of the WMGHG forcing.'

**New change in text:** 'The dominant forcing mechanism over the 1990-2015 period is changes in the well-mixed greenhouse gases (WMGHG). The global mean forcing due to $CO_2$ increased over this period by 0.66 Wm$^{-2}$ and forcing due to other WMGHG rose by 0.16 Wm$^{-2}$ (see Supplementary material for further information of the calculations). Other anthropogenic forcing mechanisms have had negligible overall changes between 1990 and 2015, though; natural forcing of volcanic eruptions and solar irradiance changes have had large changes during the period, see Prather et al. (2013) and particularly their Table AII.1.2. *In particular volcanic eruptions cause strong negative forcing on time scale of a few years. The natural forcing due to volcanic and solar irradiance changes was found to be -0.16 (-0.27 to -0.06) Wm$^{-2}$ over the period 1998-2011 (Myhre et al., 2013b).* Whereas previous studies indicated almost zero change in forcing of aerosol and ozone change this study shows by using an updated emission inventory and multi-model simulations a forcing 20% of the WMGHG forcing.'

Comment 4 - Again, you mention other studies that can be useful for interpreting the results, but do not provide a basis for quantifying how large variability is, which the reviewer had asked for. I also don't think that you should reference a paper in preparation (Sand), and it is not in the list of references.

**Response**: Quantifications in now included in the description. Sand et al. is now submitted and included properly into the reference list.

**Original change in text:** 'Differences in atmospheric abundances can be large due to different meteorological data sets (Liu et al., 2007) and surface concentrations can be influenced by interannual variation (Barnes et al., 2016), but differences associated with nudging seem to be small (Sand et al., in preparations).'

**New change in text:** 'Differences in atmospheric abundances can be large due to different meteorological data sets *(up to more than 50% in global mean aerosol burden)* (Liu et al., 2007) and surface concentrations can be influenced by interannual variation *(making 20 year trends in surface ozone due to climate variability as large as caused by changes in emissions ozone precursors)* (Barnes et al., 2016), but differences associated with nudging seem to be small *(a few percent)* (Sand et al., 2017).'

Reviewer #2

Comment 2 - This is the same statement used above, but it does not address the reviewer comment of "based on entirely different chemical and meteorological background states / years)." Please address the comment.

**Response**: We have included quantifications based on the earlier studies (see response Reviewer 1, comment 4). We feel that going beyond this is a large topic which should be left to international intercomparison efforts like AerChemMIP (CMIP6 endorsed MIP) or AeroCom.

Comment 3 - You say why you do not evaluate trends over Asia, since measurements do not go back

to 2000. But why couldn't you evaluate the present day, and not just the trend?

**Response**: This paper is about trends and in particular trends in forcing. We don't think an evaluation of the models for present day will help interpretation of the differences in forcings among the models.
5     Several of the models have been recently been evaluated over Asia (see Quennehen et al. (2016)) and repeating such an effort is of small value for this manuscript in our view.

Comment 4 - It appears that modeled trends are compared with surface observation trends, and then the trends in modeled surface and column loadings are compared. The reviewer had suggested
10    comparing with satellite-derived trends as being more directly relevant for radiative forcing. Why is that not done here?

**Response**: Trends in AOD from satellite data (or from Aeronet) is far from straightforward and needs to be documented in a proper way. In our view this is a study in itself. Work on this topic is underway within our group, but results would not be ready for a long time.
15
Comment 7 - This is the same text used to address a comment above, and I do think it clearly communicates the reason for the difference.

**Response**: See response to Reviewer 1, comment 2 which in our view help explaining the differences.

20    Comment 11 - The short response doesn't resolve the reviewer question of how methane is handled.

**Response**: We understood the question from the reviewer as a whether CH4 emissions or CH4 concentrations was used. We therefore think it is sufficient information given in the description. However, we have added that observed concentrations have been included into the models. In
25    addition, the reviewer asked if methane radiative forcings should be included in the analysis. Since our study focuses on trends in radiative forcing of short-lived species, the direct radiative forcing of methane is not included.

**Original change in text: '**In addition to the shorter-lived ozone precursors of NOx, CO, and VOC changes in the concentration of $CH_4$ is taken into account, except for the EMEP model.'

30    **New change in text: '**In addition to the shorter-lived ozone precursors of NOx, CO, and VOC changes in the *observed* concentration of $CH_4$ is taken into account, except for the EMEP model.

Reviewer #3
35
General comments - I don't see that this comment was addressed: "I would encourage the authors to emphasize the objectives of their work as well as to deepen the analysis of the process chain leading to the simulated forcing changes."

40    **Response**: We feel by adding a figure on the emissions and a new paragraph at the end of the manuscript putting the aerosol and ozone forcing in context of other radiative forcing over the 1990-2015 period that we have illustrated one important part of the manuscript and deepened the analysis. However, going beyond this and deepen the analysis further to understand model differences is a huge task. This is the goal of international efforts such as the CMIP6 endorsed MIPs AerChemMIP and
45    RFMIP. The following has been added to the introduction.

**New change in text: '***In particular, the aim is to quantify the recent changes in radiative forcing and how those compare to the values reported in the IPCC AR5.***'*

Comment 1 - The reviewer asked for a discussion of the emission changes and the arbitrary selection of models. The response addresses the emission changes but not the selection of models.

**Response**: We have added the following on the models:

**New change in text: '***Participating modelling groups are from the EU project ECLIPSE (Stohl et al. 2015) and those joining an open call for collaborating groups***.'*

Comment on Table 2 - I don't know how to interpret "and for the US periods 2000-2010 and 1990-2010 have been derived."

**Response**: The sentence has been split into two much clearer sentences.

**Original change in text: 'Models have been sampled at the grid points of the network sites and for the US periods 2000-2010 and 1990-2010 have been derived. '**

**New change in text: '***Models have been sampled at the grid points of the network sites. For the models, periods 2000-2010 and 1990-2010 have been used for comparisons with US observations.***'*

Barnes, E. A., Fiore, A. M. and Horowitz, L. W.: Detection of trends in surface ozone in the presence of climate variability, Journal of Geophysical Research: Atmospheres, 121(10), 6112-6129, 2016.

Liu, X. H., Penner, J. E., Das, B. Y., Bergmann, D., Rodriguez, J. M., Strahan, S., Wang, M. H. and Feng, Y.: Uncertainties in global aerosol simulations: Assessment using three meteorological data sets, Journal of Geophysical Research-Atmospheres, 112(D11), 2007.

Myhre, G., Shindell, D., Bréon, F.-M., Collins, W., Fuglestvedt, J., Huang, J., Koch, D., Lamarque, J.-F., Lee, D., Mendoza, B., Nakajima, T., Robock, A., Stephens, G., Takemura, T. and Zhang, H., Anthropogenic and Natural Radiative Forcing. In: Climate Change 2013: The Physical Science Basis. Contribution of Working Group I to the Fifth Assessment Report of the Intergovernmental Panel on Climate Change. T. F. Stocker, D. Qin, G.-K. Plattner, M. Tignor, S. K. Allen et al. (Editors), Cambridge University Press, Cambridge, United Kingdom and New York, NY, USA, pp. 659-740, 2013.

Prather, M., Flato, G., Friedlingstein, P., Jones, C., Lamarque, J.-F., Liao, H. and Rasch, P., IPCC 2013: Annex II: Climate System Scenario Tables. In: Climate Change 2013: The Physical Science Basis. Contribution of Working Group I to the Fifth Assessment Report of the Intergovernmental Panel on Climate Change. T. F. Stocker, D. Qin, G.-K. Plattner, M. Tignor, S. K. Allen et al. (Editors), Cambridge University Press, Cambridge, United Kingdom and New York, NY, USA, pp. 1395-1445, 2013.

Quennehen, B., Raut, J. C., Law, K. S., Daskalakis, N., Ancellet, G., Clerbaux, C., Kim, S. W., Lund, M. T., Myhre, G., Olivié, D. J. L., Safieddine, S., Skeie, R. B., Thomas, J. L., Tsyro, S., Bazureau, A., Bellouin, N., Hu, M., Kanakidou, M., Klimont, Z., Kupiainen,

K., Myriokefalitakis, S., Quaas, J., Rumbold, S. T., Schulz, M., Cherian, R., Shimizu, A., Wang, J., Yoon, S. C. and Zhu, T.: Multi-model evaluation of short-lived pollutant distributions over east Asia during summer 2008, Atmos. Chem. Phys., 16(17), 10765-10792, 2016.

[revised manuscript text omitted]